# Susceptibility and Pathology in Juvenile Atlantic Cod *Gadus morhua* to a Marine Viral Haemorrhagic Septicaemia Virus Isolated from Diseased Rainbow Trout *Oncorhynchus mykiss*

**DOI:** 10.3390/ani11123523

**Published:** 2021-12-10

**Authors:** Nina Sandlund, Renate Johansen, Ingrid U. Fiksdal, Ann Cathrine B. Einen, Ingebjørg Modahl, Britt Gjerset, Øivind Bergh

**Affiliations:** 1Institute of Marine Research, P.O. Box 1870, Nordnes, 5817 Bergen, Norway; nina.sandlund@hi.no (N.S.); ingrid.uglenes.fiksdal@hi.no (I.U.F.); Ann.Cathrine.Einen@merck.com (A.C.B.E.); 2Norwegian Veterinary Institute, P.O. Box 64, 1431 Ås, Norway; Renate.Johansen@zoetis.com (R.J.); Ingebjorg.Modahl@vetinst.no (I.M.); britt.gjerset@vetinst.no (B.G.)

**Keywords:** fish rhabdovirus, VHSV genotype III, experimental challenge, farmed fish, immunohistochemistry

## Abstract

**Simple Summary:**

VHSV (viral haemorrhagic septicaemia virus) is a globally important group of viruses, infecting a wide range of fish species, in both marine and freshwater. New initiatives are now being taken to re-establish Atlantic cod as an aquaculture species. Any susceptibility to diseases would be relevant for this industry. Atlantic cod was experimentally infected with a strain of VHSV originating from a disease outbreak of farmed rainbow trout in Norway. The fish infected by injection did develop the disease, as verified by investigations of tissue samples. However, no transmission of the virus to cohabitant fish (i.e., individuals sharing the tank with infected fish) was found. This indicates that, despite the fact that the virus is capable of causing disease in Atlantic cod, the degree and ability to infect the fish is limited.

**Abstract:**

The first known outbreak caused by a viral haemorrhagic septicaemia virus (VHSV) strain of genotype III in rainbow trout occurred in 2007 at a marine farm in Storfjorden, Norway. The source of the virus is unknown, and cod and other marine fish around the farms are suspected as a possible reservoir. The main objective of this study was to test the susceptibility of juvenile Atlantic cod to the VHSV isolate from Storfjorden. As the pathology of VHS in cod is sparsely described, an additional aim of the study was to give a histopathological description of the disease. Two separate challenge experiments were carried out, using both intra peritoneal (ip) injection and cohabitation as challenge methods. Mortality in the ip injection experiment leveled at approximately 50% three weeks post challenge. Both immunohistochemical and rRT-PCR analysis of organs sampled from diseased and surviving fish confirmed VHSV infection. No VHSV was detected in the cohabitants. The results indicate that Atlantic cod has a low natural susceptibility to this VHSV genotype III strain. One of the most extensive pathological changes was degeneration of cardiac myocytes. Immunohistochemistry confirmed that the lesions were related to VHSV. In some fish, the hematopoietic tissue of spleen and kidney showed degeneration and immunostaining, classical signs of VHS, as described in rainbow trout. Positive immunostaining of the capillaries of the gills, suggests this organ as a useful alternative when screening for VHSV.

## 1. Introduction

Viral haemorrhagic septicaemia virus (VHSV) causes viral haemorrhagic septicaemia (VHS) in a wide range of wild and farmed fish species in marine and fresh water, both in Europe, USA, and other parts of the northern hemisphere. VHSV is a member of the *Rhabdoviridae* family, genus *Novirhabdovirus*. Four genotypes, including several subtypes, have been identified. The different genotypes are mainly associated with geographical regions reviewed in [1].

In 2007, the first confirmed VHS outbreak caused by a genotype III isolate (NO-2007-50-385) occurred in rainbow trout (*Oncorhynchus mykiss*) in a fish farm in Storfjorden, Møre og Romsdal County, on the west coast of Norway. The fish showed classical signs and pathology related to VHS [2]. Prior to that, VHSV genotype III had only been isolated from wild caught marine fish around the British Isles and from farmed turbot (*Scophthalmus maximus*) reviewed in [3]. Experimental challenges with VHSV genotype III, isolated from Atlantic cod *Gadus morhua* and turbot show, low susceptibility to rainbow trout [4,5]. The reservoir of the VHSV genotype III isolate affecting the rainbow trout in Storfjorden was, and still is, unknown. However, a marine origin seems likely [6,7]. Altogether this gives indications to VHSVs potential of adaptation and virulence to new host species, which may be further increased by fish farming practice.

VHSV has been isolated from various wild-caught asymptomatic fish species, including Atlantic cod. The susceptibility of Atlantic cod to VHSV is of interest both in terms of a potential reservoir in marine fish populations, and as a risk factor affecting the Atlantic cod farming industry. A marine VHSV strain was first isolated from Atlantic cod suffering from ulcus syndrome [8]. Low prevalence of VHSV in wild caught Atlantic cod have been reported from the North Sea [9], Shetland Isles [10], the Skagerrak Sea [11], the Baltic Sea [12], and Denmark [8]. The genotype of the VHSV isolates from cod is predominantly genotype Ib, and only a few genotype III isolates are reported (www.fishpathogens.eu accessed on 7 December 2021).

The experimental susceptibility of Atlantic cod to various VHSV strains has been found relatively low, except when using injection as the challenge method [4,13,14]. However, the variable virulence of VHSV isolates to different species [15,16], and the variable susceptibility to different VHSV isolates [17,18,19] points out the necessity of testing the susceptibility and virulence of each isolate to various species in vivo, to elucidate the possible transfer of VHSV between fish species and between farmed and wild fish. At present, cod farming is a very small aquaculture industry in Norway, despite substantial attempts and investments after 2000, which were followed by setbacks, due to the financial crisis, diseases, and market problems. However, a governmentally funded national breeding program is still in place, and juveniles and adult fish are available from farmers. New initiatives are now taken to re-establish Atlantic cod as an aquaculture species [20]. In addition, capture-based cod aquaculture is a widespread practice, keeping the fish alive and feeding it after capture, making the fish available for slaughter and processing in a non-stressed state at the requested time. Thus, the susceptibility of Atlantic cod to VHSV, is once again a topic of relevance. The objectives of this study is to test the susceptibility of Atlantic cod juveniles to the marine VHSV genotype III isolate (NO-2007-50-385), from farmed rainbow trout in Storfjorden [2], in challenge experiments using ip injection and cohabitant challenge. And give a thorough description of pathology in cod suffering from VHS.

## 2. Materials and Methods

The experiments were approved by the Norwegian Food Safety Authority, as application No. (FOTS) 2011/141935. By this authorization, the experiments were judged, by the relevant national authority, to be in accordance with the WMA declaration of Helsinki.

### 2.1. Challenge Material

VHSV genotype III (NO-2007-50-385) isolated from rainbow trout during a disease outbreak in Norway in 2007 [2] was inoculated at second passage on subconfluent monolayers of BF-2-cells (ECACC, Salisbury, UK) in 1:500 dilutions in tissue culture flasks using standard techniques [21]. Culture medium was collected after 5 days incubation when full cytopathic effect (CPE) was observed. Virus titer was determined by end-point dilution and expressed as 50% tissue culture infectious dose (TCID50 mL^−1^) [22]. A 100 μL virus suspension, with titer of 10^7.3^ TCID_50_ mL^−1^ and 10^7.5^ TCID_50_ mL^−1^, were used for the ip injection and cohabitant challenge, while 100 μL BF-2 cell growth medium, containing lysed cells, was used for the control fish.

### 2.2. Experiment 1: Intra-Peritoneal Injection

#### 2.2.1. Fish and Experimental Design

Prior to challenge 15 fish were randomly sampled, using a landing net to verify that the fish were negative for VHSV. The juvenile cod, average size 24.1 g ± 5.3 standard deviation (STDV), were provided by the Institute of Marine Research (IMR) research station Parisvatnet. The fish were bath vaccinated against vibriosis, using the commercial vaccine Norwax Vibriose Marine (Europharma), 4 weeks prior to shipment to the challenge facilities at IMR, Bergen. When arriving at the challenge facilities, the cod was randomly placed, using a landing net in the respective eleven 250 L experiments tanks and acclimatized for 4 weeks until the start of the experiment. Full salinity sea water was used. The temperature was 9.4 °C at the start of the experiment, which had gradually decreased to 8.7 °C by the end of the experiment. Water flow was approximately 400 L h^−1^ (6.7 L min^−1^). The cod were fed a pellet commercial diet (Skretting). Fish groups and tanks are described in Table 1. Six tanks were used for sampling, three with 44 fish each and three with 55 fish each. Five tanks, two with 53 fish and three with 55 fish, were used to determine mortality. The number of fish in the tanks varied from 44 to 55 due to mortality during transportation and handling. The fish was starved for 24 h and sedated using metacain (Unikem) prior to challenge with intra peritoneal (ip) injection with 100 μL virus suspension with titer 10^7.3^ TCID50 mL^−1^, and thereafter rinsed for 1 h. Control fish were ip injected with 100 μL cell growth medium. The fish were monitored daily, and dead fish were removed and recorded.

#### 2.2.2. Tissue Sampling Experiment 1

Sampling times and selection of organs are described in Table 1. Three control fish were sampled at each sampling point 7, 14, 28, 84, and 155 dpc (days post challenge). The number of sampled fish from challenged fish groups is described in Table 2. Fish were randomly selected using a landing net and killed using an overdose of benzocain (Apotekproduksjon AS). In addition to the scheduled sampling clinically affected fish (fish with abnormal swimming patterns and/or external signs of disease) and recent mortalities (last few hours) were sampled throughout the experiment. Heart, spleen, and brain were divided longitudinally, one half were used for immunohistochemical examinations and the other part for PCR. All dissection equipment used was cleaned with alcohol and flame-fixed between each organ. Sampling from some individuals also included pylorus caeca and intestines, for immunohistochemical analysis only.

### 2.3. Experiment 2: Cohabitation Infection

#### 2.3.1. Fish and Experimental Design

The juvenile Atlantic cod, average size 18.5 g ± 4.1 standard deviation (STDV), were provided by IMR’s research station at Austevoll. The fish were bath vaccinated against vibriosis using the commercial vaccine Norwax Vibriose Marine (Europharma) at six and two weeks prior to shipment to the challenge facilities at IMR, Bergen. The cod was randomly placed in the respective eleven identical experiments tanks using a landing net and left to acclimatize for 3 weeks until the start of the experiment. Temperature throughout the experiment was 12 ± 0.2 °C and water flow approximately 500 L h^−1^ (8.3 L min^−1^). The experimental tanks and the diet were similar to experiment 1. Description of fish groups and tanks are given in Table 1 and Figure 1. Five tanks were used for sampling, while six tanks were used to determine mortality. A total of 70 fish were placed in each experimental tank; 50 unchallenged and 20 challenged. Challenged fish (n = 20) were ip injected with 100 µL of VHSV suspension, titer 107.5 TCID50 mL^−1^, while the control fish (n = 20) were ip injected with 100 µL cell growth medium. These 20 fish were housed in two separate, greyish coloured 17 L baskets, within the 250 L tanks, to keep them separated from the unchallenged fish (Figure 1). The side walls of the baskets were made of grids allowing free water flow between the baskets and tank, exposing the cohabitant fish to waterborne transmission.

#### 2.3.2. Tissue Sampling Experiment 2

Sampling times and selection of organs are provided in Table 1. Prior to challenge twelve fish were randomly sampled using a landing net to verify that the fish were negative for VHSV. Seven ip injected fish were sampled as positive controls to verify VHS infection. Three fish from each tank were tested for VHSV at 10 and 28 dpc and 5 fish from each tank were sampled and tested at 62 and 92 dpc. The same sampling procedure as described in experiment 1 was used. During daily monitoring clinically-affected fish and recent mortalities were sampled throughout the experiment (Table 2). No fish were dead for longer than 12 h prior to sampling and these samples were only used for PCR analysis.

### 2.4. Detection of Virus Using Real Time RT-PCR (rRT-PCR)

Samples used for PCR-detection were stored in RNAlater (Sigma-Aldrich, St. Louis, MO, USA) at 4 °C for 24 h prior to freezing at −20 °C. RNA was extracted with the automated easyMAG protocol (Biomérieux, Marcy lÈtoile, France) and measured using NanoDrop ND-1000 (NanoDrop Technologies, Wilmington, DE, USA). The rRT-PCR assay was performed in accordance with Johansen et al. [23], using primers and probe described by Matejusova et al. [24] and/or Duesund et al. [25]. Samples with cycle treshold (Ct) value ≤ 40 were considered positive. To ensure the quality of tissue samples from dead fish an internal control real-time PCR assay (CodELF) was applied to a selection of samples under the same reaction conditions as the virus assay [26].

### 2.5. Hematoxylin-Erythrosine-Saffron (HES) Staining and Immunohistochemistry (IHC)

Tissue samples were fixed in Bouin’s fixative and treated according to Evensen & Olesen [27]. Tissue sections were heated, dewaxed, and rehydrated prior to either staining with HES or IHC. IHC was performed using the monoclonal antiserum IP5B11 (provided by the National Veterinary Institute, Copenhagen, Denmark). The Vectostain^®^ universal ABC-AP kit (AK5200, Vector Laboratories Inc., Burlingame, CA, USA), including both the secondary antibody and the ABC-AP, and the DAKO Fuchsin Substrate and Chromogen system (K0625, Dako North America Inc., Carpinteria, CA, USA) were used according to manufacturers’ recommendation. The tissue sections were counterstained with Shandon’s haematoxylin and mounted in aqueous mounting medium (Aquatex, BDH laboratory, Dorset, UK). Tissue samples from the control fish were used as negative control, while VHSV positive material (provided by the National Veterinary Institute in Denmark) served as positive control. All incubations were performed at room temperature (20 °C) in a humidity chamber in a fume hood. A Leica DMBE microscope, equipped with a QImaging MicroPublisher 5 RTV camera, and QCapture software for Mac were used to capture image of the tissue sections. Tissue samples from two individuals with granulomas in kidney and one with vacuoles in brain were tested with IHC, using specific antibodies against Francisellosis [28] and Nodavirus [29], respectively.

### 2.6. Bacteriology

Swabs from head kidney and ulcers were taken from clinically affected fish during both experiments and grown in pure culture on marine agar (MA) (Difco 2216) and blood agar (BA) (nutrient agar (Oxoid) according to [30]. Bacterial isolates were identified by sequencing, as described by [31], in addition to observation of colony morphology on BA and MA.

### 2.7. Statistical Analysis Mortality Data

Since the survival and mortality data are not normally distributed, non-parametric tests were used. A 2 × 2 contingency table, performed in Statistica v 10.0 (StatSoft), was used to test for mortality differences among the treatment and control groups. As multiple independent tests were used to test differences in mortality rate among all challenged groups and control groups, a Bonferroni correction was applied to minimize the possibility of doing a type II error (Rice 1989). Experiment 1: We tested the mortality differences between the 5 fish tanks used for mortality registration and the *p* value was corrected by a factor of 5 (*p* = 0.05/5 = 0.01). Experiment 2: We tested the mortality differences between the 6 fish tanks and the *p* value was corrected by a factor of 6 (*p* = 0.05/6 = 0.0083): see Rice (1989). Yates’ correction was used since there was only 1 degree of freedom (df).

## 3. Results

### 3.1. Experiment 1 Intra-Peritoneal Injection

#### 3.1.1. Mortality

Mortality in virus injected fish groups started at 5 dpc and increased, until it declined out around 21 dpc and ended between 75 and 88% (Figure 2). During the first three weeks, mortality was also experienced in control group 2. During the first week, 15% of the fish died, and after four weeks, cumulative mortality had reached 25% (Figure 2). The mortality data from challenged fish groups found significantly different from the control fish groups are marked with red colour and the symbol “*” in Figure 2.

#### 3.1.2. Sampling

Due to high mortality rate earlier than expected the number of sampled fish was reduced from 5 to 3 fish per tank from day 14 (Table 2). Later in the experiment, 84 and 155 dpc, the number of sampled fish was increased to raise the possibility of detecting positive fish.

#### 3.1.3. Clinical Signs and Gross Pathology

The fish displayed normal swimming behavior until death occurred. At five dpc the first observations of petechiae around the fin bases, head and operculum was observed on ip injected fish. A few fish had haemorrhages around the eyes and inside the mouth cavity. Pale gills and clear ascites in addition to a pale and patched miscoloration of the spleen were frequent observations during necropsy of clinically affected and recently deceased fish. Most of the sampled fish had no stomach or intestinal content except for a yellow sticky intestinal substance. Fish that survived the acute stage of the disease (21 dpc) showed bleedings limited to the head and abdominal areas.

#### 3.1.4. Virus Detection and Tissue Distribution Using rRT-PCR

All fish sampled prior to challenge (n = 15) and control fish sampled during the experiment (n = 15) tested negative for VHSV by rRT-PCR.

Initially a selection of brain and heart samples from the first three sampling points at 7, 14, and 28 dpc (9–15 fish per sampling), were tested with two PCR assays to evaluate the diagnostic sensitivity. Assay 2 by Duesund et al. [25] showed a much higher sensitivity than assay 1 by Matejusova et al. [24] on the samples tested (Table 3). Especially in weak positive samples, the difference between the sensitivity of the two assays was noticeable. As a result, assay 2 was chosen for all further virus detection in this study.

In addition to the 92 fish sampled at the fixed sampling points, 16 clinically affected or recent mortalities were sampled and tested by rRT-PCR (Table 2). All organs, sampled from individuals between 6–55 dpc, were tested. In the first samplings (6–12 dpc) all fish were found VHSV positive. The number of positive fish then declines and only a few positive fish were detected at the end of the experiment at 155 dpc. At the last two sampling points PCR testing was limited to heart and brain samples of all fish. The remaining five organs were only tested if these tested positive. The results showed that five out of 28 individuals at 84 dpc and three out of 28 individuals at 155 dpc, tested positive (Table 2). Comparing the Ct values obtained from organs of positive individuals (data not shown), the heart was in general the strongest positive sample at all sampling points. Spleen was also found to contain high amounts of viral RNA during the first 4 weeks of the experiment. When individuals were found positive, the heart and/or brain samples were always among the organs giving positive detection for VHSV (both heart and brain samples n = 30, only heart samples n = 16, and only brain samples n = 2). No individual tested positive in the eye, gills, liver, spleen, or head kidney samples if the heart and/or brain samples tested negative.

#### 3.1.5. Histopathology and Immunohistochemistry

A total of 25 ip injected individuals were tested and compared in all organs by both IHC and PCR. All tissues with a Ct value below 33 were found positive by IHC. No positive staining with IHC was observed in tissue samples with Ct values above 33. The last verification of positively stained tissue was found in fish sampled 28 dpc.

One control fish was examined at each sampling point. No histopathology or positive immunostaining was detected in tissue samples from these fish (Figure 3A and Figure 4A).

Positive VHSV-immunostaining was seen in all tissue samples showing histopathological changes. When examining heart tissue, a general observation was degeneration of the myocardial cells (Figure 3B) and extensive VHSV-positive staining (Figure 4B). Positive staining was observed throughout the whole endocardium and myocardium of both atrium and ventricle in ten out of twelve examined hearts sampled between 6–16 dpc (Figure 4B). Focal necrosis and infiltration of inflammatory cells was observed in all 13 individuals sampled later than 28 dpc. IHC showed no detection of VHSV in the epicardium or bulbus arteriosus. Examination of heart tissue showed enlargement of endocardial cells and activation of the reticuloendothelial system (RES system), in fish testing positive for VHSV with PCR, but no positive VHSV-staining was detected in these areas of the heart (Figure 4C).

The histopathology, observed in the spleen, varied from no detection (n = 17) to various degrees of necrosis and degeneration of both white and red pulpa with immunostaining (n = 8) (Figure 3C and Figure 4D). The positive staining varied accordingly, from extensive staining of most of the hematopoietic tissue in fish sampled 6–7 dpc (n = 5) (Figure 4D) to sparse staining of single cells 12–14 dpc (n = 5).

Positive staining of head kidney tissue samples was also variable. Extensive staining of most of the hematopoietic tissue was observed in samples taken at 6–7 dpc (n = 5) (Figure 5E). Later, at 12–14 dpc, only sparse staining of single cells was observed in two out of five examined head kidneys. It should also be noted that the two kidney samples with observed granulomas tested negative for Francisella by IHC.

Another trend was that in organs with less positive staining, such as liver, gills, and eye, the positive staining was associated with endothelial cells in blood vessels and capillaries (Figure 4F–I). Positive staining of circulatory cells (leucocytes) was also observed in the eye (n = 2) (Figure 4J), gills (n = 2) and brain (n = 1). In addition, positive staining of the iris was observed (n = 1) (Figure 4K).

On one occasion, the intestine was included in the organ material sampled for histological examination (16 dpc). Degeneration of muscle cells in muscularis circularis and muscularis transversum was observed (Figure 3D), and this corresponded well with positive immunostaining (Figure 4L).

Positive staining in the brain was weak and difficult to observe. Twenty-five brains were analysed with both IHC and PCR method. Of these, only four out of 17 PCR-positive brains were confirmed positive for VHSV with IHC (7, 14, and 28 dpc). The positive staining was observed in the metencephalon, both in single neuron cells (Figure 4M) and foci of cells (Figure 4N). Some vacuoles were observed in the individual illustrated in Figure 4N. The brain tissue was tested also for nodavirus infection by IHC and found negative.

### 3.2. Experiment 2 Cohabitation Infection

#### 3.2.1. Mortality

The majority of ip injected fish died between three and six dpc (Figure 5). By 14 dpc, they were all diseased, and the baskets, in which they were placed, were removed from the tanks. Accumulated mortality % of both ip injected and cohabitating fish are shown in Figure 3. The mortality data from cohabiting fish groups found significantly different from the control fish groups are marked with red colour in Figure 5.

#### 3.2.2. Clinical Signs and Gross Pathology

Post-mortem examinations of the virus injected fish revealed similar findings to the challenged fish in experiment 1. Cohabitant fish, sampled during the regular samplings, did not show any signs of pathology or disease. Post-mortem examination of diseased cohabitant fish showed petechiae on the abdomen, around the fin base, and head. Pale gills were found on a few individuals. In addition, ulcers on the caudal peduncle were seen in most individuals. Erosion of fins, especially the caudal fin, was also frequently observed.

#### 3.2.3. Virus Detection and Tissue Distribution Using rRT-PCR

All fish sampled prior to challenge (n =12) and control fish sampled during the experiment (n = 22) tested negative for VHSV by rRT-PCR. The ip injected fish (n = 7) were tested and found positive for VHS-virus. None of the cohabitant fish (n = 48) tested positive for VHSV in any organ samples (gills, heart, spleen, head kidney, and brain).

The CodELF PCR assay gave stable Ct values within the range of 12–15.

#### 3.2.4. Histopathology and Immunohistochemistry

Fish injected with virus showed pathology and positive immunostaining with anti-VHSV serum similar to findings in experiment 1 (n = 6). None of the cohabitant fish tested positive with anti-VHSV serum or showed any signs of pathology related to VHS infection (n = 16).

### 3.3. Bacteriology

Bacterial analysis was performed on diseased fish during both experiments to possibly explain mortality not related to VHS. Experiment 1: Bacterial isolates sampled from ulcers (n = 19) and head kidney (n = 12) of seven fish were sequenced. The 16S rRNA sequencing analysis revealed a mixed bacterial micro flora of *Vibrio* spp. and *Tenacibaculum* spp. from ulcers, while *Vibrio* spp. and *Photobacterium* spp. were identified from the kidneys. Experiment 2: A mixed bacterial flora of *Vibrio* spp. (n = 8) was isolated from ulcers of two individuals. No bacterial growth was obtained from bacterial swabs taken from head kidney of four individuals. Bacterial sampling did not provide sufficient data to conclude whether or not some fish died of bacterial causes.

## 4. Discussion

Our results show that Atlantic cod are susceptible to VHSV genotype III isolate (NO-2007-50-385) when the virus is injected into the fish. Increased mortality and classical pathological signs, such as degeneration of the hematopoietic tissue, were observed, along with more unique changes in the heart. The virus was, however, not horizontally transferred in our cohabitation experiment, which indicates that Atlantic cod has a low susceptibility to this VHSV isolate at this study’s given conditions.

Although experimental studies have shown that Atlantic cod is not highly susceptible to VHSV [4,13,14], the virus have been isolated from wild-caught Atlantic cod during screening surveys [11,12], including genotype III isolates [13]. This means Atlantic cod are naturally exposed and susceptible to VHSV to some degree. In the ip injection study VHSV was detected in most examined organs during the acute stage of the disease (6–20 dpc), and pathological examinations confirmed the development of VHS. The frequency of positive fish and organs then declined, and only a few positive fish were detected at 84 and 155 dpc. This is comparable to challenge experiments with both rainbow trout and fathead minnow *Pimephales promelas*, in which presence of VHSV was not detectable in tissue samples towards the end of the experiments (74 days post infection) [32] and experiment with Japanese flounder *Paralichthys olivaceus* testing negative at 50 dpc [33]. This could indicate that the fish is able to clear itself of the virus infection or that the virus resides in specific organs in low amounts. Even longer challenge trials than 155 days are needed to look into this matter, and stress at the end of the experiments could possibly reveal hidden infections.

Our results show that when examining Atlantic cod challenged by ip injection the heart and/or brain were always among the positive organs in individuals testing positive for VHSV by rRT-PCR. No individuals tested positive in the eyes, gills, liver, spleen, or head kidney samples if the heart and/or brain tested negative. The highest frequency of PCR positive fish was found in heart samples (Table 3). This corresponds well with the observed histopathology (Figure 4B and Figure 5B,C). The extensive positive IHC staining and histopathological changes observed in the myocardium is similar to findings in turbot *Scophthalmus maximus* [19], freshwater drum *Aplodinotus grunniens* [34], and muskellunge *Esox masquinongy* [35]. The heart, as an important target organ for VHSV, has also been suggested in Japanese flounder *Paralichthys olivaceus* [33]. Iida et al. [33] detected the highest titres of VHSV in the heart, kidney, and spleen, and positive heart samples reappeared after performing a stress test 50 dpc. This indicates that heart may be one of the best organs for detection of VHSV in clinically healthy carrier fish, and this fits well with our results. Viral load and tissue distribution of VHSV, in various species, is only partly known and further knowledge are needed to reveal where the virus is residing in clinically healthy fish of different species [19,32,35,36,37,38,39,40].

In the present study the histopathological examinations of heart samples revealed activation of the RES system in the endocardium (Figure 4D). The RES system in the endocardium is known for its ability of endocytotic properties and detoxification capabilities [41]. The large surface of the endocardium facilitates contact with the bloodstream. Activation of the RES system as result of a viral infection is, therefore, likely, and our histopathological results support this.

Brain samples frequently tested positive for VHSV with PCR assays; however, the IHC staining in brain tissue was scarce, weak, difficult to observe, and limited to single cells. In general, results from IHC studies may, to some extent, depend on the location of the tissue section and, in particular, tissues with few positive cells. Careful analyses of several serial sections of such tissue samples may limit this inaccuracy. Our results are in contrast to the histological findings by Dale et al. [2], which describe the inflammation, necrosis, and extensive positive staining in affected rainbow trout brains, after challenges with the current genotype III VHSV isolate. VHSV detection in brain and meninges has also been described from several fish species, both in fresh and salt water [34,35,42]. The results from Lovy et al. [42] show that VHSV can persist in nervous tissue in Pacific herring, *Clupea palasii*, after the virus has been cleared from other tissues.

Positive immunostaining of gill tissue was observed in the endothelial cells of capillaries (Figure 4G,H) and not in the external surface, indicating an active infection of the gill tissue and not just virus particles attached to the external mucus surface of the lamella. Similar findings were reported in turbot [19], Pacific herring [40,42], and rainbow trout [32]. High prevalence of VHSV has been detected by rRT-PCR in gill tissue from Norwegian spring-spawning herring *Clupea harengus* [23]. More recently, Hershberger et al. [40] suggested that VHSV persists in association with the gills in Pacific herring that has recovered from VHS, as long-term shedding of the virus seems to be connected to gill epithelial cells. In the light of the findings listed above, gills may be a useful organ, if the aim of the screening is to detect distribution of VHSV *per se*. Not only for their water filtering function, but also as an organ in which VHSV seems to persist post infection. In addition, gills are also a relatively large organ and easy to sample.

The positive staining of circulatory cells and endothelia of blood vessels was a frequent and consistent observation in several tissues, including the gills, eye, and liver (Figure 4F–J). Vasculitis was reported as the most consistent and severe form of lesions, present in most tissues of freshwater drum suffering from VHS) (more specifically, the necrosis of endothelial cells [34]). This indicates that receptors for VHSV are present both on endothelial cells and blood cells. With increasing knowledge, regarding viral load and tissue distribution of VHSV, sampling protocols may need to be optimised to each fish species.

The ip injection experiment in this study with VHSV resulted in a cumulative mortality rate between 40–55% approximately 3 weeks post challenge. An ip injection experiment on salmon *Salmo salar* using the same genotype III isolate NO-2007-50-385 resulted in a cumulative mortality of 52% around day 20 post infection [2]. The challenge dose used in the salmon experiment, 10^5^ TCID_50_ injected in each fish, was lower than in this study (10^7.3^ TCID_50_ mL^−1^ and 10^7.5^ TCID_50_ mL^−1^), suggesting that salmon is slightly more susceptible to this genotype III isolate than Atlantic cod.

The cohabitation challenge in this study did not result in a VHS infection in cohabiting fish. This is comparable to previous studies with cod [4]. The amount of virus shedded from the ip injected fish in experiment 2 into the water is not known. However, it is likely that there was a relatively stable shedding of virus into the environment during the first week following infection, when most of the injected fish developed disease and died, as has been previously shown in similar experiments with other VHSV isolates [40,43]. Thus, the results indicate that Atlantic cod is not highly susceptible to VHSV isolate NO-2007-50-385 unless ip injection is used as transmission route. Cohabitation experiments have been used in other fish species with different VHSV isolates providing mortality rates, which show that shedding of virus and horizontal transmission, under experimental conditions, is possible [15,43,44,45,46,47]. Bath challenge with high amounts of virus is a possibility for future challenge trials and additional stress may be needed to make cod more susceptible. Younger fish and even yolk sac larvae are also something that needs to be tested.

VHSV is widely distributed on the Northern hemisphere and isolated from an increasing number of fish species. VHSV genotype IVb has caused disease and mortality to several wild fish species inhabiting in the Great Lakes of North America [35,48,49,50]. In addition, there are indications of transmission of VHSV of genotype IVa from wild fish, such as herring, to farmed Atlantic salmon in the British Colombia area [51]. This points to the potential of spreading and emergence of virus in new host species. Atlantic cod is exposed [11,12], and to some degree susceptible, to VHSV, including genotype III isolates [13]. The impact of VHSV infection in Atlantic cod is, therefore, of interest, especially in the light of the recent initiatives to re-establish Atlantic cod as an aquaculture species. Oral transmission through feed is a possible route of entry that could be further examined. In the wild, cod prey on various VHSV susceptible fish species, such as herring [23,52], which might be a source of infection. Oral transmission, through ingestion of VHSV infected trout *Salmo trutta*, has been experimentally shown for the northern pike *Esox lucius* [53] and rainbow trout being fed fish pellets containing VHSV [54]. Cod were, however, not found susceptible after being fed VHSV-containing pellets [14]. It is possible that Atlantic cod may serve as an asymptomatic carrier of VHSV. Special circumstances, such as age and weakened immune system (caused by stress or secondary infections), may be needed, in order for Atlantic cod to become susceptible to VHSV and/or develop disease. This could possibly explain the fact that cod shows low susceptibility to VHSV in challenge trials.

## 5. Conclusions

Our results demonstrate that Atlantic cod is susceptible to the VHSV genotype III isolate from rainbow trout in Norway 2007, following intraperitoneal injection, but no horizontal transfer of infection to cohabitants was found. This indicate that, despite findings of VHSV in wild cod and the positive findings in the i.p injected fish, VHSV genotype III is not highly pathogenic to the species. Atlantic cod can still be carriers of the virus and spread the disease to other fish species. Even so, it is important to keep in mind that VHSV is a virus with a great variety of genotypes and host specificity, and that other genotypes may have different patterns, regarding host range and virulence.

The findings also show that the gills are affected by VHSV, and gills may be suitable for sampling and screening for the virus.

## Figures and Tables

**Figure 1 animals-11-03523-f001:**
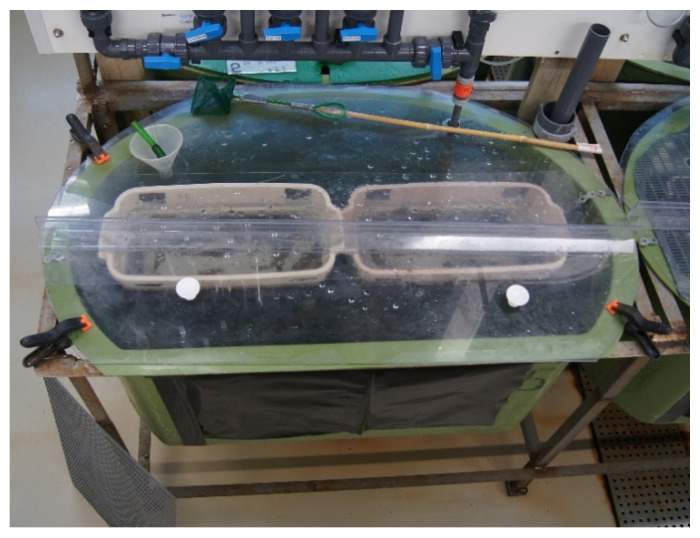
Photo of one challenge unit, one tank and two baskets, used in experiment 2. The 50 cohabitant fish were kept in the green 250 L tank, while the 20 VHSV ip infected fish were housed in the two greyish coloured 17 L baskets. The side walls of the baskets are made of grids allowing free water flow between the baskets and tank exposing the cohabitant fish to waterborne transmission.

**Figure 2 animals-11-03523-f002:**
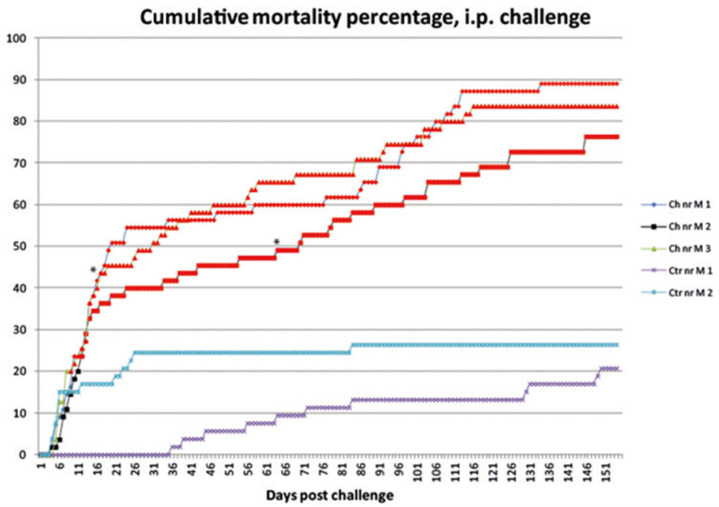
Cumulative mortality percentage of challenge (Ch nr) (n = 55 in each tank) and control (Ctr nr) (n = 53 in each tank) Atlantic cod (*Gadus morhua*) fish groups in experiment 1. M = fish groups used for mortality registrations only. Challenged groups are juvenile cod, ip injected with, respectively, 100 μL virus suspension, containing VHSV isolate NO-2007-50-385, TCID_50_ 10^7.3^ mL^−1^. Control fish were ip challenged with, respectively, 100 μL BF-2 cell growth medium, containing lysed cells. Mortality data from challenged fish groups found significantly different from control group 1 are marked with a red colour (*p* < 0.01). * = from that day onwards; mortality data from challenged fish groups were also significantly different from control group 2 (*p* < 0.01, Bonferroni correction).

**Figure 3 animals-11-03523-f003:**
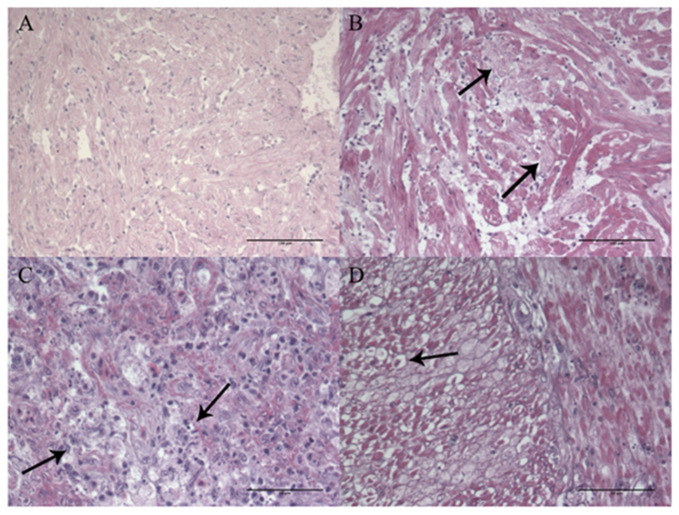
(**A**–**D**). Hematoxylin-erythrosine-saffron (HES) staining of paraffin sections from Atlantic cod (*Gadus morhua*) juveniles. (**A**) Heart from control fish 14 dpc. Scale bar 100 µm. (**B**) Heart of fish sampled 7 days dpc, showing pathology and degeneration of the myocardium (Arrows). Scale bar 100 µm. (**C**) Spleen from fish sampled 7 dpc, showing severe necrosis and degeneration of hematopoietic tissue (Arrows). Scale bar 50 µm. (**D**) Intestine of fish sampled 16 dpc, showing necrosis in muscularis circularis and muscularis transversum (Arrow). Scale bar 50 µm.

**Figure 4 animals-11-03523-f004:**
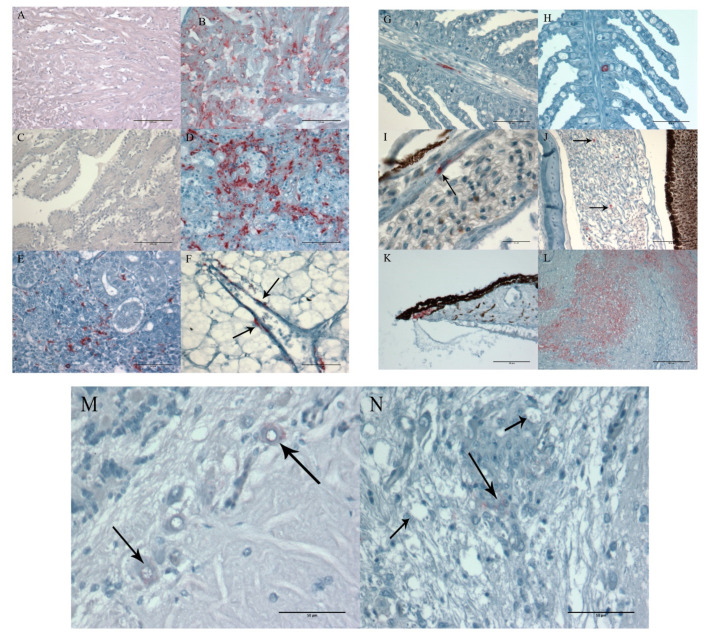
(**A**–**N**) Immunohistochemistry (IHC) (red) staining of paraffin sections of various organs from Atlantic cod (*Gadus morhua*) juveniles, using the monoclonal anti-VHSV IP5B11 serum, ABC-AP enzyme system and visualised with fuchsin. Positive IHC is visualised by red colouring of tissue. Shandon haematoxylin was used as counterstaining, giving the tissue different tones of blue. (**A**) Heart from control fish sampled 14 days post challenge (dpc), with no positive staining. Scale bar 100 µm. (**B**) Heart from same individual as in Figure 4B, showing positive immunostaining in myocardium and endocardium. Notice the lack of positive staining of the epicardium. Scale bar 50 µm. (**C**) Showing activation of the reticuloendothelial system (RES), 28 dpc. Scale bar 100 µm. (**D**) Spleen of same individual as in Figure 4C, showing positive staining of the hematopoietic tissue. Scale bar 50 µm. (**E**) Head kidney sampled 6 dpc showing positive immunostaining in the interstitium. Scale bar 50 µm. (**F**) Liver sampled 6 dpc showing positive staining of cells associated with blood vessel (arrows). Scale bar 50 µm. (**G**) Gills from fish sampled 6 dpc showing positively stained cells associated with blood vessel in the primary lamellae. Scale bar 50 µm. (**H**) Gills from recently died fish 12 dpc, showing positive staining aligning the endothelia of the capillary in the secondary lamellae. Scale bar 50 µm. (**I**) Corpus choroidale from an eye, sampled 7 dpc, showing positive endothelial-like cells (Arrow). Scale bar 15 µm. (**J**) Corpus choroidale from an eye sampled 14 dpc with positively stained circulating cell (leukocyte-like cells) in the capillary system (arrows). Scale bar 50 µm. (**K**) Positive immunostaining of iris from recently died fish 16 dpc. Scale bar 50 µm. (**L**) Intestine of same individual as in Figure 4D showing positive immunostaining in muscularis circularis and muscularis transversum. Scale bar 100 µm. (**M**) Brain from fish 28 dpc showing positive staining of neurons in metencephalon (Arrows). Scale bar 50 µm. (**N**) Brain from fish 28 dpc showing metencephalon with a cluster of inflammatory-like cells with some positive staining (large arrow). Vacuoles are shown using small arrows. Scale bar 50 µm.

**Figure 5 animals-11-03523-f005:**
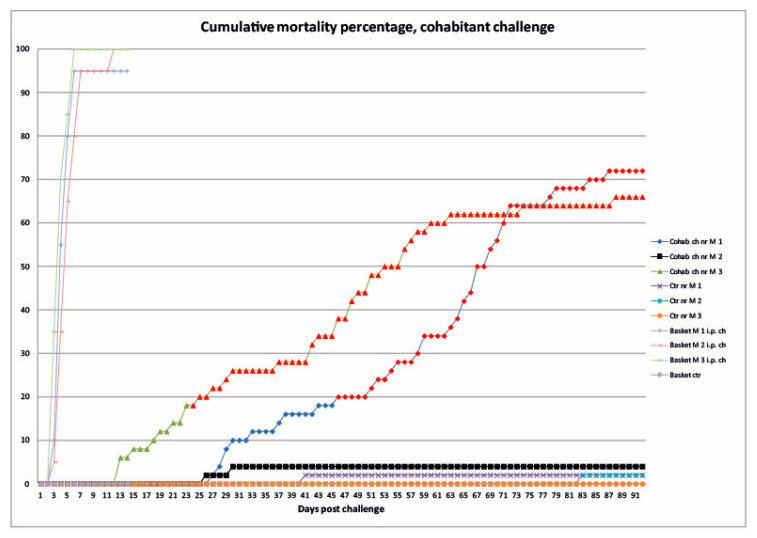
Cumulative mortality percentage of Atlantic cod (*Gadus morhua*) fish groups included in experiment 2; cohabitation challenge fish groups (Cohab ch nr) (n = 50 cohabitants in each tank); control fish groups (ctr nr) (n = 50 cohabitants in each tank); M = fish groups used for mortality registrations only. The challenged fish groups are juvenile cod, exposed to VHS isolate NO-2007-50-385, through cohabitation. Virus injected fish was kept in separate baskets within the fish tanks, referred to as “Basket M 1–3 i.p. ch” (n = 20 ip injected fish in each tank). Mortality data, from challenged fish, groups found significantly different from control groups are marked with a red colour (*p* < 0.083, Bonferroni correction).

**Table 1 animals-11-03523-t001:** Experimental design of experiment 1 and 2. ip = intra peritoneal, cohab = cohabitant fish, Ctr = control group, Ch = challenge group, M = tanks used for mortality registrations, S = tanks used for sampling only, dpc = days post challenge.

Experiment 1; i.p. Challenge				Experiment 2; Cohabitation Challenge			
Fish group and tanks	Nr of fish	Sampling dpc	Sampled orgns	Fish group and tanks	Nr of cohab + i.p. injected fish	Sampling, dpc	Sampled organs
Ctr nr M1-2	53		Mortality registration only	Ctr nr M1-3	50 + 20		Mortality registration only
Ch nr M1-3	55			Ch nr M1-3	50 + 20		
Ctr nr S1-3	44	7, 14, 28, 84, 155	Eyes, Gills, Heart, Liver, Spleen, Head kidney, Brain	Ctr nr S1-2	50 + 20	10, 28, 62, 92	Eyes, Gills, Liver, Spleen, Head Kidney, Brain, Pylorus caeca, Intestine
Ch nr S1-3	55			Ch nr S1-3	50 + 20		

**Table 2 animals-11-03523-t002:** Experiment 1, intra peritoneal injection: Number of positive VHSV detections by rRT-PCR. Organ samples from five fixed sampling points (7, 14, 28, 84, 155 dpc) in addition to sampling of clinically-affected fish and resent mortalities (*). Dpc = days post challenge.

No. of Positive Organs									Comments
Dpc	No. of Sampled Individuals	Gills	Heart	Spleen	Kidney	Eye	Brain	Liver	
6 *	7	7	7	7	7	7	7	7	All organs sampled from individual between 6–55 dpc were tested
7	15	14	15	15	14	9	11	13	
12 *	1	1	1	1	1	1	1	1	
14	12	6	9	4	4	3	7	4	
14 *	3	3	3	3	3	3	3	3	
20 *	1	1	1	1	1	1	1	1	
28	9	1	5	1	1	2	2	0	
44 *	2	0	1	0	0	0	0	0	
55 *	2	0	1	0	0	1	1	0	
84	28	0 of 5	4	0 of 5	1 of 5	2 of 5	1	0 of 5	Five individuals tested positive in either brain or heart. These 5 were tested in all organs
155	28	0 of 3	2	0 of 3	0 of 3	1 of 3	1	0 of 3	Three individuals tested positive in either brain or heart. These 3 were tested in all organs

**Table 3 animals-11-03523-t003:** Initial comparison of two PCR assays results on brain and heart samples from Atlantic cod *Gadus morhua* 7–28 dpc with VHSV genotype III. Number of rRT-PCR VHSV-positive organs/number tested. Assay 1 [24], versus assay 2 [25]. dpc = days post challenge.

Days Post Hatching	Assay 1		Assay 2	
dpc	Brain	Heart	Brain	Heart
7	3/15	10/15	11/15	15/15
14	1/9	3/9	4/9	7/9
28	1/9	1/9	2/9	5/9

## Data Availability

All data can be available upon request.

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
