# Peer review of "Susceptibility and Pathology in Juvenile Atlantic Cod Gadus morhua to a Marine Viral Haemorrhagic Septicaemia Virus Isolated from Diseased Rainbow Trout Oncorhynchus mykiss"

_animals, 2021, doi:10.3390/ani11123523_

Round 1

Reviewer 1 Report

Comments in the attached file

Author Response

On behalf of all authors, I thank the reviewer for his/her efforts. I have edited the linguistics of the manuscript, hopefully improving the English to an acceptable level. 

Reviewer 2 Report

The Animals-1472157 manuscript describes the susceptibility to genotype III of the VHS virus in Atlantic cod. The paper is extremely interesting and well written: the authors used an excellent experimental design, well detailed.

The introduction is well articulated and comprehensive; the M&M are well described. In the chapter of the experimental design, both in experiment 1 and in experiment 2, the authors should insert in the text the number of fish used, better describing the number per tank and better specify the choice of those numbers, this to facilitate the reading of the text without having to decipher the attached tables that can mislead the reader; it is also important to clearly insert the number of replicas for each experimental thesis.

The results are well exposed and clear. The discussion is well developed and usable. On line 539 delete the date in brackets and insert the relative citation (33) at the point on line 543.

For these reasons, in my opinion, the manuscript can be published after a minor revision which only implies an implementation in the text of data present in the tables but not very usable.

Author Response

On behalf of the authors, I thank the Editor and Referees for speedy and professional processing of our manuscript. I have edited the manuscript and made changes according to the reviews.

I have proofread the manuscript more, and done minor changes which are purely linguistic, but should improve the English.

A more substantial change has been done, according to suggestions buy one Referee:

Line 116 onwards has been changed to:

«Fish groups and tanks are described in Table 1. Six tanks were used for sampling, three with 44 fish each and three with 55 fish each.  Five tanks, two with 53 fish and three with 55 fish, were used to determine mortality. The number of fish in the tanks varied from 44 to 55 due to mortality during transportation and handling»

In 2.3.1 this has already been taken into account, so this text is unchanged.

«Five tanks were used for sampling while six tanks were used to determine mortality. A total of 70 fish were placed in each experimental tank; 50 unchallenged and 20 challenged.»

We have not done any changes with the Table texts or the Tables itself.

The reference with date (33) at line 539 in the first submitted version has been changed according to the Referee´s report.